# Data-Augmented Hybrid Named Entity Recognition for Disaster Management by Transfer Learning

**Hung-Kai Kung [1], Chun-Mo Hsieh [2], Cheng-Yu Ho [1], Yun-Cheng Tsai [3],\* and Hao-Yung Chan [4] and Meng-Han Tsai [4]**

1. Department of Geography, National Taiwan University, Taipei 10617, Taiwan; b06208001@ntu.edu.tw (H.-K.K.); b06208030@ntu.edu.tw (C.-Y.H.)
2. Department of Economics, National Taiwan University, Taipei 10617, Taiwan; b06303059@ntu.edu.tw
3. School of Big Data Management, Soochow University, Taipei 111002, Taiwan
4. Department of Civil and Construction Engineering, National Taiwan University of Science and Technology, Taipei 10607, Taiwan; d10705005@mail.ntust.edu.tw (H.-Y.C.); menghan@mail.ntust.edu.tw (M.-H.T.)
* Correspondence: pecutsai@gm.scu.edu.tw

**Abstract:** This research aims to build a Mandarin named entity recognition (NER) module using transfer learning to facilitate damage information gathering and analysis in disaster management. The hybrid NER approach proposed in this research includes three modules: (1) data augmentation, which constructs a concise data set for disaster management; (2) reference model, which utilizes the bidirectional long short-term memory–conditional random field framework to implement NER; and (3) the augmented model built by integrating the first two modules via cross-domain transfer with disparate label sets. Through the combination of established rules and learned sentence patterns, the hybrid approach performs well in NER tasks for disaster management and recognizes unfamiliar words successfully. This research applied the proposed NER module to disaster management. In the application, we favorably handled the NER tasks of our related work and achieved our desired outcomes. Through proper transfer, the results of this work can be extended to other fields and consequently bring valuable advantages in diverse applications.

**Keywords:** damage information gathering; disaster management; data augmentation; transfer learning; named entity recognition; chatbot

## 1. Introduction

### 1.1. Demand for Natural Language Processing (NLP) in Disaster Management

This research stems from the difficulties encountered in keyword recognition of the conversation-based system proposed in our related work [1]. The system is a chatbot developed for supporting school building inspection tasks, and the main contribution is the process improvement of questions analysis and information retrieval. Nevertheless, one of the significant inconveniences is that the system is incapable of grasping keywords from the feedback of school building safety inspections. Analyzing data from plain texts to obtain in-depth information and conduct further explanatory analysis is difficult. In other words, the messages of the chatbot left by assessors presented in lists. Therefore, an efficient analysis method is still absent in the context of mixed information. A large amount of artificial screening and processing is required to compile a concise report from chaotic feedback quickly.

In disaster management, there are several studies that applied machine learning and natural language processing (NLP) [2,3]. One of its subdomains is named entity recognition (NER), which helps

extract words from sentences. With the assistance of this advanced technique, this research targets ensuring the quality of the message collected through the conversation module and further analyzing those results. That is to say, we can make sure that we do collect the information we need rather than unrelated messages, and thus analysis for disaster management is possible. Of the many NER solutions, one of the simplest ones purely based on predefined "rules". However, rule-based methods' performance is unsatisfactory, especially for systems dealing with sentences that involve foreign words. Besides, these methods require rules to be maintained and updated to retain their performance over time. Hence, the current research explores other advanced NER models as a superior solution to perform NER tasks in disaster management.

### 1.2. Drawbacks of General NER Models

With the use of NER techniques, keywords can now extract from the reviews of school building safety inspections. However, the results are not appropriate because most NER models can only achieve high performance in general cases. For specific domains, even the most advanced models may not yield the intended results without adequate specialization. For example, poor model performance may stem from the absence of desired labels and the insufficiency of training data in a given domain. CKIP CoreNLP, one of the latest models established by Academia Sinica, Institute of Information Science, can only tackle general tasks even though it is one of the most profound models for completing Mandarin NER tasks [4,5]. In sum, the results of this model become useless and uninterpretable when the situation narrows down to a specific domain, such as disaster management. Therefore, general NER models that apply to disaster management are necessary.

### 1.3. Model Revision for Disaster Management

The module is initially developed for related work [1]. Our related work (The Human-Computer Interaction Lab, 417 Department of Civil and Construction Engineering, NTUST) is mainly a chatbot to collect the post-disaster data and feedback. With those data in hand, we can analyze the data and provide disaster managers with compact information. However, when there are more and more data and feedback provided by the assessors, the analyzing work must be tough and inefficient without the assistance of NLP techniques. Hence, with the module of this paper, the back-end can input the raw text into the module and get the parsing results. In that way, the displaying module in the related work can get better and efficient information for the manager. Take another example in the chatbot, according to the question in the examination. The conversation module of the related work (chatbot script) has expected to get the "accurate" answer. Without the assistance of the module, the chatbot relies on merely a rule-based method to check if the assessor provides "enough" information. Hence, the result may be flawed and not accurate at all. However, with the module, the conversation module can enhance the overall quality of data collected.

The current research aims to recognize named entities in damage information gathering and analysis tasks in disaster management. However, without sufficient reviews of safety inspections, a reliable NER model is challenging to build. Hence, this research utilizes transfer learning, which is one of the most popular techniques in machine learning. With the assistance of transfer learning, a NER model can build based on existing models with minimal training data. As this NER model designed to target disaster management, the model should recognize specific terms; that is, the model needs further modification through data augmentation. After the appropriate enhancement, the NER model can recognize not only familiar Mandarin entities but also jargons in the domain.

## 2. Literature Review

### 2.1. Review of Related Work

In our related work, we developed a conversation-based system for school building safety inspections [1]. The system designed to notify users, gather data and present an informative front-end

dashboard in the field of disaster management. This system is mainly a chatbot integrated with all the functions and modules mentioned above. With the full availability of communication devices and smartphones, communication software has developed rapidly [6,7]. LINE is a messaging application that has outperformed others by exceeding standards and offering the most diverse features [8]. Hence, we chose LINE as the communication and data transmission platform in our related work. We supplemented it with various message templates and finally developed a LINE-based chatbot for school building safety inspection for disaster management [9]. However, the feedback gathered from the chatbot merely comprised plain texts, which lacked depth and efficiency. The non-analytical and unsystematic data confined the related work to narrow and inflexible processes. For the conversation module, the conversation processes restricted to finite scripts, and the chatbot sometimes failed to respond to users precisely. For the display module, the disorganized feedback led to inefficiency in the front-end dashboard. For example, a report from the chatbot may include lines of description rather than a concise and explanatory layout detailing the events (what caused the damage), products (what were damaged), locations, and the rest. Hence, our proposed chatbot will benefit from NLP techniques for analyzing texts and offering in-depth information.

## 2.2. Evolution of NER

With years of training in natural languages, humans can easily and intuitively distinguish named entities; the same is not valid for machines [10]. Earlier NER techniques were mostly rule-based approaches. By collecting relevant data extensively to build an extensive dictionary database, rule-based approaches are solutions with relatively few technical barriers. In our previous work, we attempted to adopt a rule-based method and found that only named entities covered by the dictionary extracted successfully. Although the performance of rule-based approaches could improve by extending the grammatical rules, they still lack robustness and portability [10]. Whenever a new corpus appears, the rules need to be modified to maintain the best performance; this requirement increases the maintenance cost accordingly [11]. In practice, dialogues come in various combinations, and new named entities emerge over time. Marking these named entities is not realistic manually. In other words, building a time-invariant dictionary that contains all named entities is not feasible [10]. As a solution to the shortcomings of purely rule-based solutions, association rule mining developed. Association rule mining establishes recognition rules based on the attributes of target entities and the characteristics of neighbors [12]. However, this method still performs poorly outside the rules.

With the advancement of technology and the improvement of computation, recognizing named entities is now possible with machine learning. Machine learning-based NER transforms a given problem into classification tasks. When we use statistical models and algorithms, machine learning-based NER methods attempt to find patterns and relationships in sentences and identify the named entities [10]. Such techniques have adopted to address some problems in disaster management [13]. They have also aided decision-makers in retrieving static documents and real-time information by integrating semantic and temporal types of terms into NER tasks to achieve improved questions analysis performance. However, existing studies that used such techniques focused on information retrieval in limited target documents; such a process is different from damage information gathering and analysis, which involves data collection and analysis. Besides, these studies did not maximize the NER techniques. Named entities that did not cover in training data can still recognize according to sentence patterns and contexts by adopting advanced machine learning techniques into NER problems. By maintaining a memory based on historical information, recurrent neural network (RNN) models can predict current output on long-distance features. For sequence tagging tasks, long short-term memory (LSTM) models heavily used. The network of the LSTM resembles the framework of RNNs. The crucial difference is that in capturing long-range dependencies, the LSTM substitutes the purpose-built memory cells with the hidden layer updates in the RNN. Also, the LSTM can easily access past and future input features for a specific time during sequence tagging [14]. Through the use of bidirectional LSTM (Bi-LSTM) networks and the forward and backward states

for two-way passing, the omissions caused by relying on only one direction resolved. Meanwhile, the conditional random field (CRF) focuses on the sentence level instead of individual positions and is capable of producing high tagging accuracy in general. Relative to other algorithms, Bi-LSTM–CRF limits its dependency on word embedding [14]. When introducing Bi-LSTM–CRF to other models, word embeddings need not be organized further, and specific training data can import directly.

## 2.3. Introduction to Transfer Learning

All models of NER are for improving accuracy and portability. In NER's field, the application of a model across different domains has always been a considerable challenge [10]. Supervised learning involves building statistical models from labeled training data. However, if the training data are insufficient, the model cannot perform well due to the sparseness of the data [10]. As the official dictionary for formal terms and the existing reviews of school building safety inspection for the current research are inadequate to build a brand-new NER model, we attempt to introduce transfer learning, which is one of the most popular frontiers of machine learning, to solve the issue of data shortage.

Since 1995, transfer learning has attracted the attention of scholars around the world under different names [15]. For example, multitask learning attempts to learn multiple target tasks simultaneously and find out the function that benefits all tasks [15,16]. Nowadays, the source task's roles and the target task in transfer learning are no longer symmetrical; transfer learning focuses on target tasks instead of learning all tasks at the same time. In other words, under the current definition, transfer learning aims to learn from one or more original tasks and apply the result to the target tasks. Based on the existing model of NER, transfer learning is applied in three different levels [17]. (1) Cross-domain transfer: Under the premise of the same language, the adequately named entities of each field are mere "sublanguages." They still share the patterns and characteristics of the language. Cross-domain transfer learning can divide into two types: label remapping and disparate label sets. (2) Cross-application transfer: On the premise of the same language, named entities still share the same primary language characteristics even if their applications differ. The model architecture is the same as the previous one with disparate label sets. (3) Cross-lingual transfer: Transfer learning across languages mostly relies on additional packages, and the performance is sensitive to package size and quality [18]. Therefore, cross-language transfer, which involves the sharing of character embedding and character-level neural networks, is proposed for similar languages [17]. In the current research, we implement the NER module via the cross-application as mentioned above transfer, that is, the cross-domain transfer with disparate label sets. Figure 1 shows the model architecture.

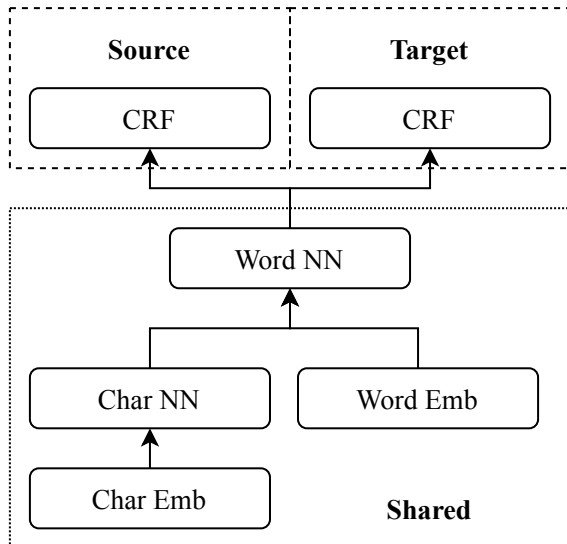

**Figure 1.** Cross-Domain transfer with disparate label sets and Cross-Application transfer [17].

*2.4. The Implementation of Hybrid NER*

Aside from familiar Mandarin entities, the model requires enhancement to recognize the sentence patterns and specific terms in the domain. It is proven that integrating maximum entropy model (MaxEnt) or hidden Markov model (HMM) with handcrafted grammatical rules enhances NER's performance [19]. The MaxEnt or hidden Markov model (HMM) comes with handcrafted grammatical rules to enhance NER's performance. The combination of a rule system and a machine learning model is called hybrid NER [10]. Such an approach recognizes the characteristics of sentence patterns through machine learning and utilizes the appended training data and labels to improve its capability in practice. Compared with general NER models, hybrid NER recognizes named entities in a specific field more accurately [10].

## 3. Methodology

This research aims to construct an augmented NER model that goes beyond the reference model. The augmented NER model specializes in recognizing keywords in the field of damage information gathering and analysis in the context of disaster management through transfer learning. The model consists of three main modules. The first module, data augmentation, establishes the compact database of named entities and specific patterns regarding disaster management from the official dictionary for formal terms and from existing inspection reviews, representing the primary target of interest in this work. This module can further split into three tasks: NER, pattern specification, and vocabulary update. The second module, the reference model, utilizes Bi-LSTM–CRF as an approach to general Mandarin NER and serves as the third module's foundation. The third module, the augmented model, implements cross-domain transfer techniques with disparate label sets to reinforce NER for damage information gathering and analysis in disaster management [17]. Specifically, it combines data augmentation and transfers learning based on a reference model.

Figure 2 illustrated the interaction between the three modules. Initially, this research sets up a reference model as the basis of the augmented model. Afterward, the predetermined corpora generated through the data augmentation module added to the augmented model. Subsequently, the augmented model applies transfer learning and integrates the results of the other two modules. After several iterations of training, the augmented model is finally feasible to use in NER for damage information gathering and analysis in disaster management.

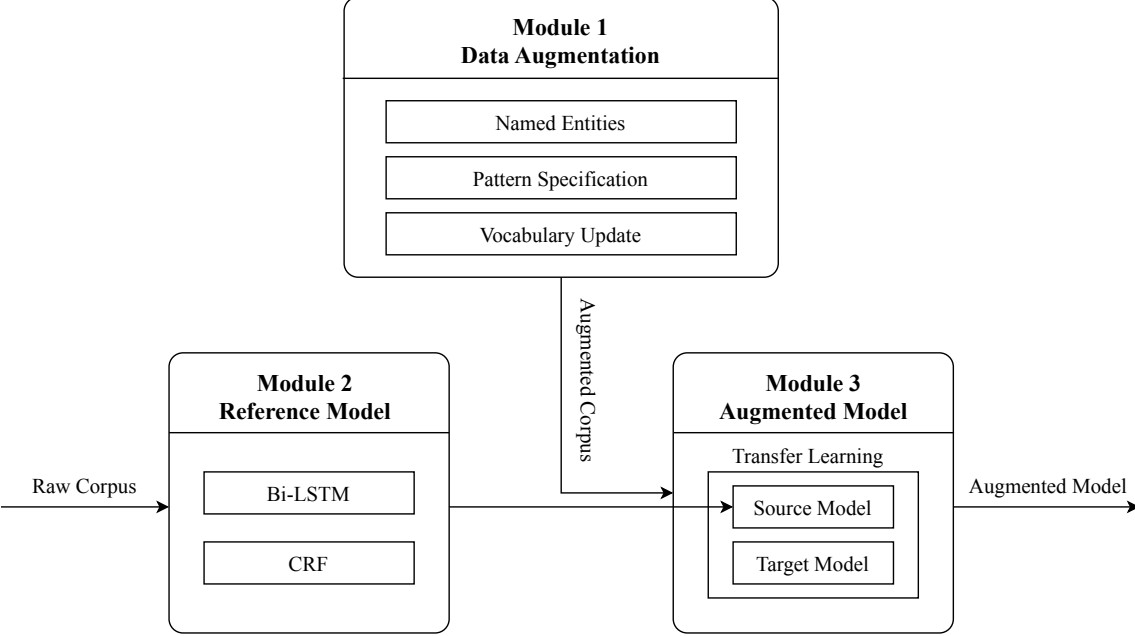

**Figure 2.** The interaction between 3 modules.

*3.1. Data Augmentation*

As particular structures, sentence patterns, and specific terms in disaster management limit the available data for training, this research must perform data augmentation. For texts used in messaging applications, the formations usually differ from those in large scales, such as newspapers and articles. Furthermore, specialized databases are required to recognize specific jargon in particular domains. In other words, the performance of a general NER model may flaw by the unfamiliar vocabulary used in specific domains and, thus, imprecise recognition. Hence, before building the target model, this research constructs a concise database to strengthen the training process concerning damage information gathering and analysis in disaster management. As a result of the advancement in machine learning, establishing a comprehensive database from scratch is no longer necessary. The data augmentation module can decompose into three tasks: NER, pattern specification, and vocabulary update. Along with these three tasks, the module provides the predetermined training data required for later use and enables the augmented model to perform NER tasks in the target domain.

3.1.1. Named Entities

For the specific vocabulary about school building safety inspections and disaster management, this research collects raw data mainly from the following:

1.  National Science and Technology Center for Disaster Reduction (NCDR), which is the official dictionary for formal terms in Taiwan; and
2.  The historical messages regarding school building safety inspections from the associated LINE chatbot proposed in our related work contain the most frequently used terms, colloquial usages, and abbreviations.

The former's data allow the NER model to recognize professional terms while the latter benefit the model by effectively parsing the common usage of regular people and augmenting the former's potential insufficiency. The database used for general NER inherited from the Institute of Automation, Chinese Academy of Sciences, CASIA [20]. Based on this database, this research then enhances and expands the categories or tags and the records of each tag.

3.1.2. Pattern Specification

Unlike the sentences frequently used in people's daily lives, the sentences used in disaster management sometimes go beyond the ubiquitous corpora. Thus far, the existing general NER models based on standard corpora and large-scale texts often perform poorly in target tasks, especially colloquial sentences. Therefore, this research analyzes and introduces the historical message data from the associated LINE chatbot developed in our related work. Table 1 enumerates the introduced patterns (Only the entities list in Appendix A has the entity labels. Otherwise, the buffer words such as, Start, End, Action, etc are labeled as O.). With these patterns, the module generates training data by replacing target entities with other words. The other buffer words, such as greetings, adjectives, and action words, are also introduced to the training sentences while excluding the buffer words that do not match the target entities.

**Table 1.** Inducted Patterns.

| Sentence Pattern |
| --- |
| Start + <Product> + <Event> + End |
| Start + <Location> + prep + <Product> + <Event> + End |
| Action + <Location> + prep + <Product> |
| Start + <Location> + <Product> + <Event> + Action |
| . . . |

### 3.1.3. Vocabulary Update

Owing to the ampleness of words, one cannot realistically and practically build a time-invariant database that records all vocabularies. This research aims to build a database that is sufficient for model training to solve this problem. During its validation and online operation, the model naturally encounters foreign words that do not record. Nevertheless, the model can predict and capture newly encountered entities according to sentence contexts. The captured new words can store in a temporary space. After the frequency of these words reaches the threshold or by manually reviewing them, they added to the database; that is, the new words recursively strengthen the model's performance.

### 3.2. Reference Model

The reference model adapted from the general Mandarin NER model developed by the open-source [20]. This module is also the source of the augmented model. This module is constructed through RNNs and employs Bi-LSTM–CRF to recognize and fetch relevant entities in input sentences. The mechanism of the module can divide into two layers, namely, Bi-LSTM and CRF.

### 3.2.1. Layer 1: Bidirectional Long Short-Term Memory, Bi-LSTM

The first layer of the reference model predicts the possible entity-tag of each Mandarin Appendix A. After that, the layer takes the highest possible score of the entity tags to predict each Mandarin character. Moreover, Bi-LSTM computes by reading the input sentence with logical order not only recursively but also reversely. Relative to the traditional LSTM, Bi-LSTM performs with high accuracy by acknowledging the past and the future input features within sentences [21]. Hence, it addresses the drawback of the traditional LSTM that may "forget" the previous section of a sentence and result in potential inconsistencies [14]. The processing by Layer 1 yields the primitive output comprising all characters within a sentence that tagged with corresponding entity tags. Figure 3 illustrated the mechanism.

However, the first output is merely the set of individual optimizations for all characters, and the rough combination may result in nonsensical sets of named entities, as indicated in Figure 3. In other words, taking only Layer 1 into account may lead to the fallacy of composition. That is, the set of individual optimizations may not reflect the overall optimization. Hence, the model uses Layer 2 to tackle this imperfection of Layer 1.

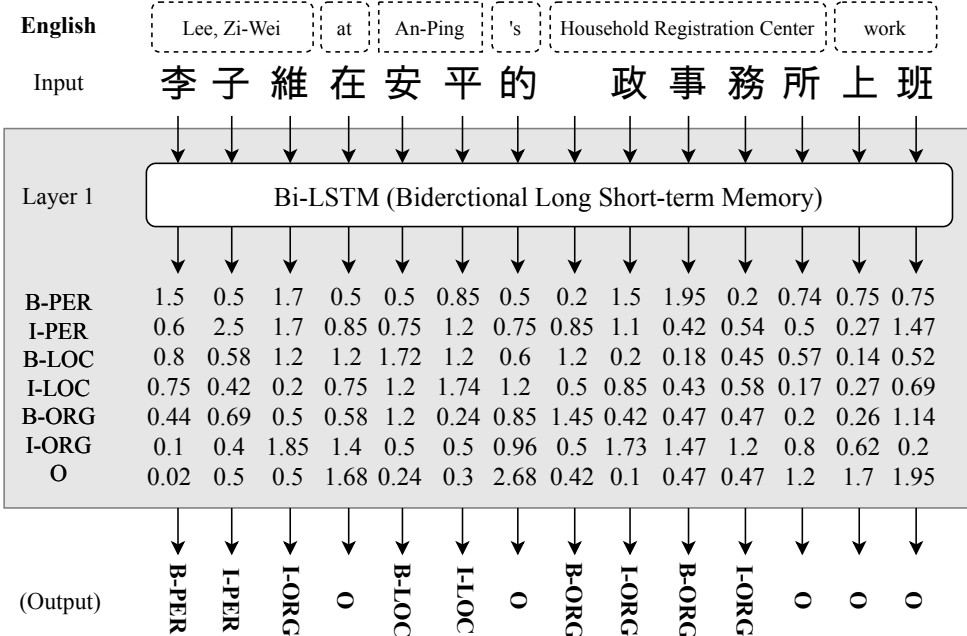

**Figure 3.** The Bi-LSTM layer. (Output) is the mid-product of Bi-LSTM.

### 3.2.2. Layer 2: Conditional Random Field, CRF

The second layer, the CRF, performs NER by learning from Mandarin sentence patterns. That is, the CRF examines whether or not the first output of Layer 1 meets the grammar and common usages. Once it realizes that the first output consists of inaccurate or rare entity tags, the model revises the prediction sets until the output is grammatically correct. In other words, the CRF clarifies the primitive output by the unit of the word and examines whether or not the prediction accords with general sentence structures. By screening out abnormal primitive outputs, Layer 2 decreases the prediction errors from the sole use of Layer 1 and results in enhanced NER performance. Figure 4 illustrated the integrated mechanism of the two layers.

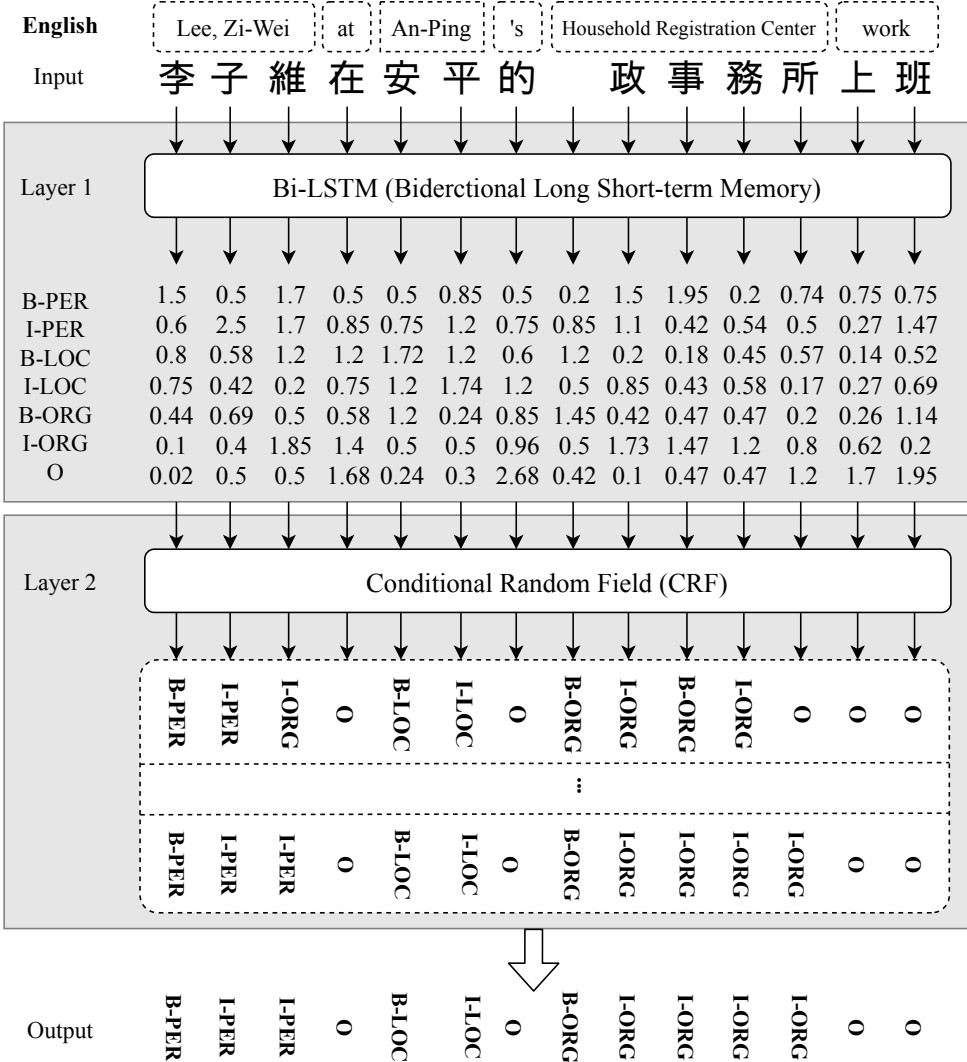

**Figure 4.** The integration of two layers.

In summary, this module implements Bi-LSTM–CRF for general NER. The sophisticated processes include the following:

1.  Reading of input sentence,
2.  Splitting of words into characters,
3.  Layer 1: Bi-LSTM, results in a series of entity-tag predictions,
4.  Layer 2: CRF, screens and revises the primitive outcome of Layer 1,
5.  Comparison of the results of the previous process with tagged entities to minimize losses recursively.

At this point, the named entities captured by the reference model are merely general entities. In an attempt to recognize the specific terms of disaster management, the following module, the augmented model, utilizes transfer learning to match the target entity tags from the source tags of the reference model.

### 3.3. Augmented Model

The previous module, the reference model, achieves the preliminary recognition of Mandarin sentences and words. However, to recognize messages pertain to disaster management, this research would have to build a comprehensive and enormous training set to train the reference model if transfer learning is not adopted. Unfortunately, as mentioned in Section 1.3, the lack of adequate corpora regarding disaster management hinders the development of a NER model from scratch. Therefore, this research utilizes transfer learning to address the difficulty of constructing a specialized NER model. The augmented model is constructed based on the reference model. With the introduction of transfer learning to the existing Bi-LSTM–CRF, this module merges the resources from the data augmentation module by updating the reference model's parameters. One of the solutions to transfer between source tasks and target tasks is label remapping whenever feasible. The mechanism underlying label remapping involves defining a new label set for a target task and mapping the original label set from the source task to the new one individually. However, no one-to-one relationship exists between source tags and target tags in this research. Hence, the augmented model realizes the process through a cross-domain transfer with disparate label sets [17].

Figure 5 illustrated the structure of the augmented model. Referring to the flowchart, this module first initializes the source and target models whose main structures inherited from the reference model which is labeled as 1. The two submodels share the same augmented training data from the data augmentation module but with disparate label sets, hence the initialization of parameters are separate labeled as 2. Subsequently, the training process enters to several times of epoch iterations which is labeled as 3. Each epoch iteration parses the whole training data into different sub-training sets to execute a batch iteration. In each batch iteration, which is labeled as 4, the two submodels in the augmented model are trained through Bi-LSTM together, but they undergo distinct CRF processes separately concerning the corresponding label sets for their tasks. Meanwhile, in the process of model optimization, the two submodels have the same shared parameters but with different specific parameters. The former is jointly optimized for the two submodels and updated after each batch iteration. The latter is extended from the former and is optimized and updated, depending on their own label sets. After completing all the batch iterations, an epoch iteration accomplished. The augmented model is constructed after several epoch iterations and can finally perform NER tasks in disaster management.

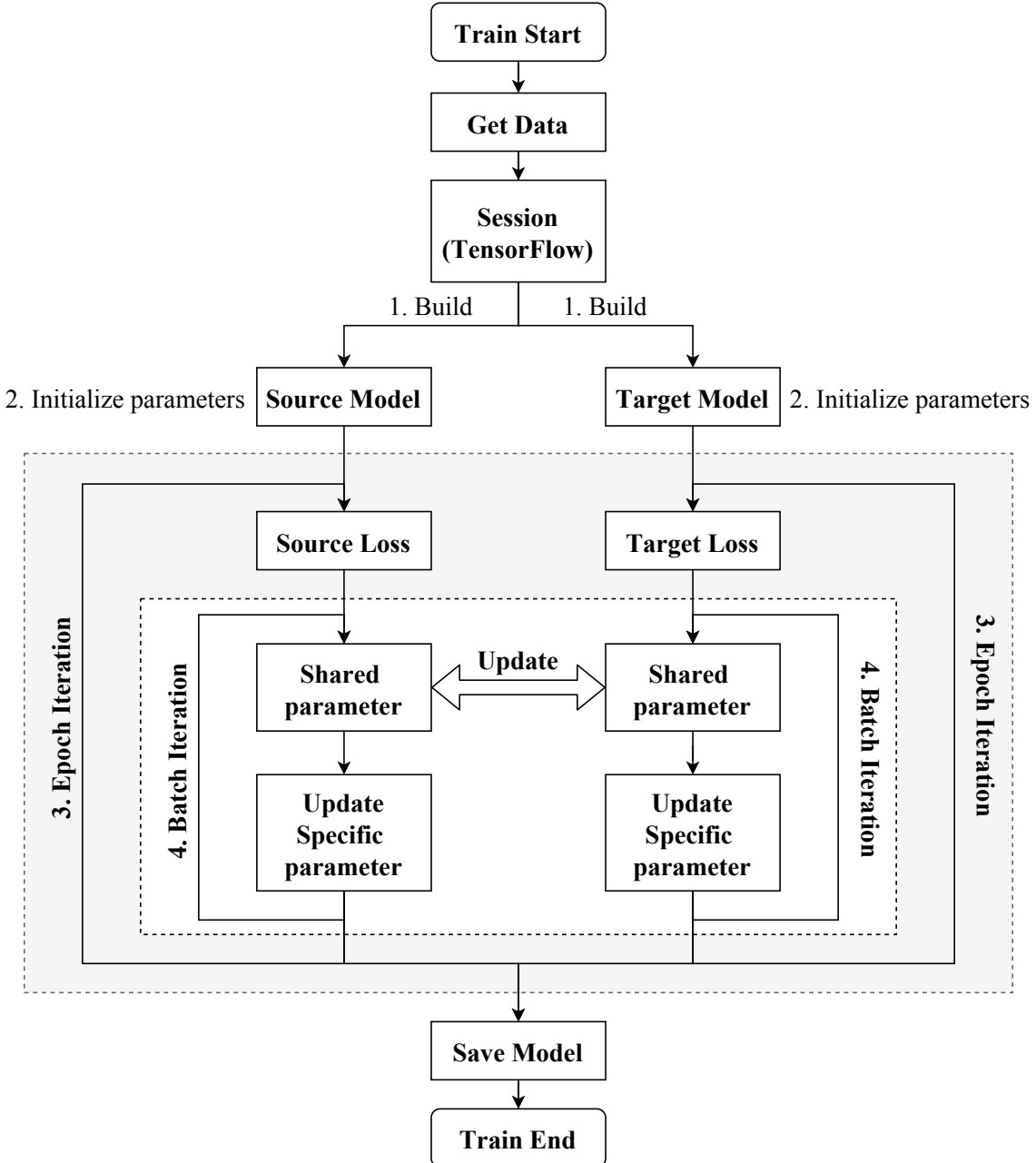

**Figure 5.** The Structure of the Third Module.

## 4. Results

### 4.1. Performance of Augmented Model

Table 2 presented the NER performance of the augmented model toward damage information gathering and analysis in disaster management after 40 epoch iterations. In the process of data augmentation, the training data now include large numbers of proper nouns and sentence patterns. After the two-layered Bi-LSTM–CRF processing, the entities of interest inside sentences can be recognized. The descriptive statistics shown in Table 2 indicate that except for the entity tag of "organization names", which obtains a relatively low score, all the other tags possess about 75% to 95% precision. Given such performance of the augmented model, its implementation in our application for damage information gathering and analysis in disaster management (i.e., school building safety inspections in our related work) shows great promise.

**Table 2.** Model Training Result with Evaluation.

|  | **Precision** | **Recall** | **FB1** |
|---|---|---|---|
| Location | 93.33% | 94.30% | 93.81% |
| product_name | 87.90% | 90.58% | 89.22% |
| person_name | 70.25% | 78.83% | 74.29% |
| time | 77.90% | 74.48% | 76.15% |
| Event | 70.72% | 70.30% | 70.51% |
| Org_name | 59.62% | 44.29% | 50.82% |

*4.2. Recognition of Named Entities*

The augmented model is a hybrid NER approach that combines handcrafted grammar rules through data augmentation and Bi-LSTM–CRF from the reference model. Compared with purely rule-based NER, the hybrid NER does not require a well-built database that records all potential outcomes. The augmented model can infer named entities from sentence patterns because of the built-in Bi-LSTM–CRF framework. Accordingly, in the case of foreign words, this approach is still a feasible solution.

To demonstrate how the augmented model recognizes named entities, the following illustrates two cases. In the first case, all named entities are available in the database. In the second case, some named entities do not record in the database. These cases illustrated in Figures 6 and 7.

In Case 1, the sentence "Ceiling in Pu-Tong Building 307 cracks seriously" gives a clear statement. The module can fully recognize the entities within the sentence. Thus, the model pairs "product_names" with "ceiling", "location", with "Pu-Tong Building 307", and "event" with "cracks seriously".

In Case 2, the sentence "The tiles and steel structure of the College of Social Sciences Building 308 damaged severely" states the problem generally. Unlike in Case 1, this sentence consists of two subjects, a general description and a set of unknown product names. The result given by the model shows that both subjects, "tiles" and "steel structure", are recognized as "product_name" despite being unknown to the module. The location "College of Social Sciences Building 308" is also recognized successfully, and the general response "damaged severely" is classified as "event".

The two cases above illustrate the process of the augmented model and how it copes with unknown entities. Through these cases, we can understand the ability of the augmented model and how it functions. With the hybrid NER approach, the recognition process that requires much human labor can now accomplish itself. With the surge of diverse report messages, our solution expected to be flexible and promising in information gathering and analysis in disaster management.

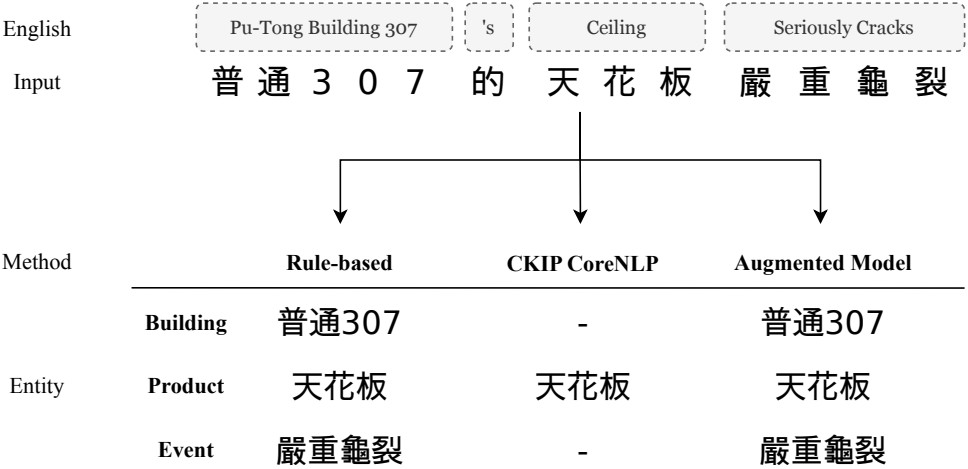

**Figure 6.** Case1: all name-entities are included in database.

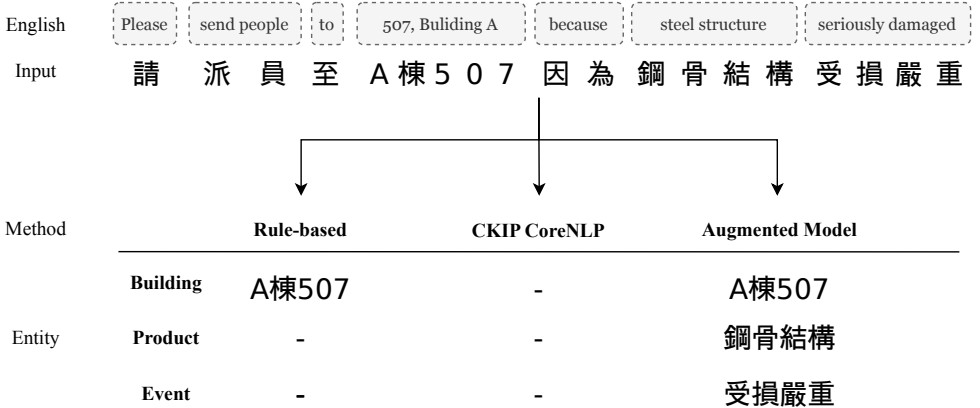

**Figure 7.** Case2: some name-entities are not included in database.

### 4.3. Vocabulary Database Update

In addition to performing NER based on sentence contexts, this research also considers the ability to update the database with new words. When the augmented model cannot extract the correct entities from the sentences, the unknown words added into the prepared list. After the frequency of those words reaches the threshold or through manual examination, the database includes them.

Take "Cracks appear on the wall", for instance. Based on past experiences, messages that users provide should include at least two types of entities, such as "product_name" and "event" to report the situation comprehensively. In the given an example, the entities are "wall" and "Cracks". If the model fails to acquire one of the two entities, users might have used unknown words or made typographical errors. In this case, the unknown word is "Cracks". Hence, the unknown entity "Cracks" appends to the prepared list as a candidate to add to the database. When the appearance of "Cracks" reaches a pre-established threshold, the database is automatically updated. Furthermore, the candidates in the prepared list can add through manual confirmation. Through this mechanism, the database can update. When the words show up again, the module is now familiar with them and is prepared to recognize them.

### 4.4. Interfacing with Messaging Apps

In its implementation in the LINE chatbot from our related work, the NER module follows the same logic described when handling user messages. The module of the research takes only "text" as inputs. Nevertheless, ordinary mobile devices such as smartphones, iPads, etc support voice input by transforming voice messages into texts. Thus, the Chatbot can receive the processed messages as appropriate inputs for the module. Upon receiving a report message sent by a user, the module attempts to extract any recognizable entities. When it fails to achieve recognition, thereby resulting in a message that lacks integrity, the chatbot asks for further details, as shown in Figures 8 and 9. In this scenario, the same vocabulary database update strategy described previously works. With the introduction of the hybrid NER model, the procedure of school building safety inspections improved, and the chatbot performs efficiently and intelligently. The vocabulary database is also automatically updated. In general, this procedural improvement facilitates data collection, and its application can further expand into other fields after proper transfer.

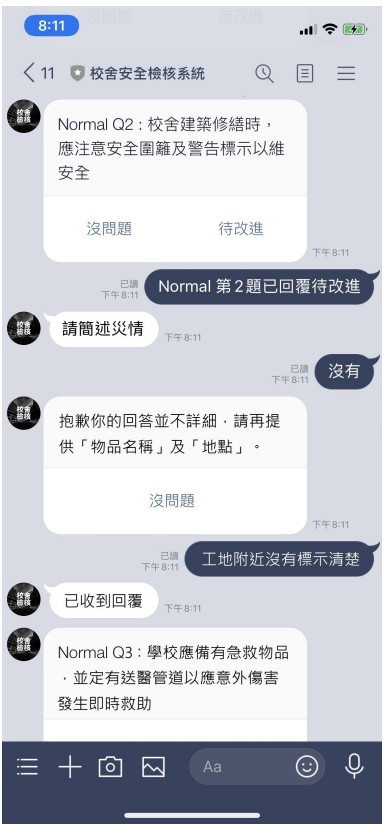

**Figure 8.** NER requests the user for details—Mandarin.

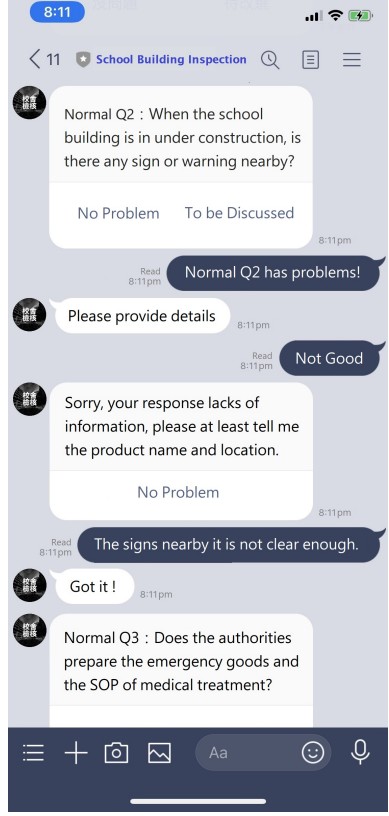

**Figure 9.** NER requests the user for details—English.

## 5. Discussion

### 5.1. Contribution

This research utilizes the existing general Mandarin NER model, that is, transfer learning, and applies data augmentation to construct an augmented NER model specifically for the disaster management domain. With the hybrid NER approach, the named entities that cannot recognize in rule-based and general NER models can now identify. As described in Section 4.4, the integration of the proposed model with the LINE chatbot from our previous work is practical and indeed enhances the procedure of data collection. In conclusion, this work provides an augmented NER model designed for disaster management. This model is a public, open-source project that is available on Github: https://github.com/MorrisHsieh3059/NERiana.

### 5.2. Improvement in Application

According to the model performance in Section 4.1, the augmented model adapts well to the application. In contrast to rule-based NER, the hybrid NER makes the whole recognition process flexible. In other words, the unknown entities can identify, and the vocabulary database grows automatically. In the long run, the hybrid NER solution is an effective way to recognize the messages reported by users. Through transfer learning, adapting the general NER model to the disaster management domain is possible. Hence, the solution effectively deals with the disadvantage of the general NER model in specific domains. In conclusion, with the combination of data augmentation, the general NER model, and transfer learning, the hybrid NER approach is more flexible than purely rule-based methods, and its implementation in the disaster management field is more suitable than that of a general NER method.

### 5.3. Limitation

Applying the augmented model to the disaster management field is beneficial. Through the augmented model, the procedure of data collection of the associated LINE chatbot now yields flexibility. However, by adopting NER, the research inevitably inherits its limitations. NER is a knowledge-intensive task, and attaching external knowledge to the model is crucial when applying NER approaches [22]. That is, the support of the domain knowledge is critical when using NER in a specific field. The following shows several cases involving NER models aided by external knowledge in many domains other than disaster management. Transfer learning is applied in biomedical named entity recognition to link the model to the biomedical domain [23], and the HMM-based approach of machine learning used to make the model suitable for any language domain [24]. With data augmentation, NER models can fit in the specific domains that researchers are interested in and thus gain powerful performance. These examples show that, on the one hand, augmented NER models designed for distinct domains do provide sturdy solutions relative to general NER models. On the other hand, these augmented models work exclusively. The augmented models customized for specific domains lack external validity, and any requested models for other domains require domain-related data retraining. Hence, the augmented model of the current research is constrained to the realm of disaster management. Although it functions well in this field, it cannot tackle any other NER tasks beyond this scope.

### 5.4. Future Work

This research employs NER to extract keywords for damage information gathering and analysis in the disaster management field. However, existing conversation processes merely carried out by responding to observed keywords rather than genuinely understanding the users. Thus, the chatbot merely performs an analysis based on entities instead of understanding, for example, the urgency or seriousness conveyed behind the failure to notify authorities, resulting in acute emergencies promptly. This research expects to achieve natural language understanding (NLU) in the future. With NLU, this research can promote fluency during conversation processes and extend the flexibility

of the chatbot's response. Furthermore, the hybrid NER approach presented in this research inherits the general NER's ability and the domain knowledge acquired from transfer learning. With the advancement of these techniques, the augmented model fits in the domain of disaster management and performs well in the application of our related work. Following the same idea, similar models can implement in domains outside disaster management. The proposed hybrid approach expected to facilitate the development of other models and bring diverse application benefits.

## 6. Conclusions

This research successfully established a hybrid NER approach to handle NER tasks in disaster management. The data gathering procedure from our related work [1], a conversation-based system for school building safety inspections, benefits from the proposed approach. The first module, data augmentation, constructs a compact but robust dataset augmented to the field of disaster management. The second module, the reference model, utilizes the Bi-LSTM–CRF structure to recognize named entities in the sentences reported. The third module, the augmented model, applies transfer learning to build the label set and construct the NER model specific for disaster management based on the two previous modules. With the help of these modules, gathering reviews from the messaging app in our related work improved. The chatbot now has the superior ability to absorb implicit sentence patterns and thus employed in practice. The proposed hybrid NER approach designed for the field of disaster management. By substituting the first module's data and altering the desired label set in the third module, we can make this research compatible with other particular domains.

**Author Contributions:** Conceptualization, Y.-C.T., H.-K.K., C.-M.H. and C.-Y.H.; data curation, H.-K.K., C.-M.H. and C.-Y.H.; formal analysis, C.-M.H., H.-K.K. and C.-Y.H.; funding acquisition, M.-H.T.; investigation, C.-Y.H., H.-K.K. and C.-M.H.; methodology, C.-M.H., H.-K.K. and C.-Y.H.; project administration, Y.-C.T.; resources, Y.-C.T. and H.-Y.C.; software, C.-M.H., H.-K.K. and C.-Y.H.; supervision, Y.-C.T.; validation, H.-K.K., C.-M.H. and C.-Y.H.; visualization, C.-M.H.; writing—original draft, H.-K.K., C.-M.H. and C.-Y.H.; Writing—review & editing, Y.-C.T., H.-Y.C. and M.-H.T. All authors have read and agreed to the published version of the manuscript.

**Funding:** This work was financially supported by the Taiwan Building Technology Center from The Featured Areas Research Center Program within the framework of the Higher Education Sprout Project by the Ministry of Education in Taiwan. Additionally, Hung-Kai Kung and Yun-Cheng Tsai are supported in part by the Ministry of Science and Technology of Taiwan under grant 108-2813-C-031-079-M.

**Acknowledgments:** The authors are appreciated and grateful to the Human-Computer Interaction Lab, Department of Civil and Construction Engineering, NTUST, for the enlightenment and support of the research.

**Conflicts of Interest:** The authors declare no conflict of interest.

## Abbreviations

The following abbreviations are used in this manuscript:

| | |
|---|---|
| Bi-LSTM | Bidirectional Long Short-term Memory |
| BNER | Biomedical Named Entity Recognition |
| CASIA | Institute of Automation, Chinese Academy of Sciences |
| CRF | Conditional Random1 Field |
| HMM | Hidden Markov Model |
| NER | Named Entity Recognition |
| NLP | Natural Language Process |
| NLU | Natural Language Understanding |
| NTUST | National Taiwan University of Science and Technology |
| RNNs | Recurrent Neural Networks |

## Appendix A. Tags of the Label Set in the Augmented Model

O                   Other, non-specific entity
B-product-name      The beginning character of a product entity
I-product-name      The non-beginning character of a product entity
B-time              The beginning character of a time entity
I-time              The non-beginning character of a time entity
B-person-name       The beginning character of a person entity
I-person-name       The non-beginning character of a person entity
B-org-name          The beginning character of a organization entity
I-org-name          The non-beginning character of a organization entity
B-company-name      The beginning character of a company entity
I-company-name      The non-beginning character of a company entity
B-location          The beginning character of a location entity
I-location          The non-beginning character of a location entity
B-event             The beginning character of a event entity
I-event             The non-beginning character of a event entity

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
