# Peer review of "Data-Augmented Hybrid Named Entity Recognition for Disaster Management by Transfer Learning"

_applsci, doi:10.3390/app10124234_

Round 1
Reviewer 1 Report
This is an interesting paper that develops a NER for disaster management. It seems to be used as a chatbot. Could it be used with voice messages? In what conditions?
Can you describe how the developed system could be used in a real-life situation?
What is the difference the results presented in this paper compared to the ones in “Question-Answering Dialogue System for Emergency Operations”, obviously, besides the fact that this paper targets disaster situation.
Line 19: What is “school9 building”? Or is it just a typo?
Line 210: What do you mean by: “The reference model adapted from the general Mandarin NER model developed by the open-source ? ].”?
There seems to be some missing references in text (see lines 140, 190, 211, 219, 221, 364, etc.)
Is it possible to generalize the algorithm for other languages, too? What would have to change?

Author Response
- This is an interesting paper that develops a NER for disaster management. It seems to be used as a chatbot. Could it be used with voice messages? In what conditions?
Ans.
The module of the research takes only ‘text’ as inputs. Nevertheless, ordinary mobile devices such as smartphones, iPads, etc. support voice input by transforming voice messages into texts. Thus, the Chatbot can receive the processed messages as appropriate inputs for the module.
- Can you describe how the developed system could be used in a real-life situation?
Ans.
The module is initially developed for our related work [3]. Our related work (The Human-Computer Interaction Lab, 417 Department of Civil and Construction Engineering, NTUST) is mainly a chatbot to collect the post-disaster data and feedback. With those data in hand, we can analyze the data and provide disaster managers with compact information. However, when there are more and more data and feedback provided by the assessors, the analyzing work must be tough and inefficient without the assistance of NLP techniques. Hence, with the module of this paper, the back end can input the raw text into the module and get the parsing results. In that way, the displaying module in the related work can get better and efficient information for the manager. Take another example in the chatbot, according to the question in the examination. The conversation module of the related work (chatbot script) has expected to get the ‘accurate’ answer. Without the assistance of the module, the chatbot relies on merely a rule-based method to check if the assessor provides ‘enough’ information. Hence, the result may be flawed and not accurate at all. However, with the module, the conversation module can enhance the overall quality of data collected.
- What is the difference the results presented in this paper compared to the ones in “Question-Answering Dialogue System for Emergency Operations”, obviously, besides the fact that this paper targets disaster situation?
Ans.
In our related paper, the main contribution is the process improvement of gathering data [3]. However, the major target of this paper is to ensure the quality of the message collected through the conversation module and further analyze those results. In this paper, we adapt NER to extract those important keywords from the reports, and thus we can make sure that we do collect the information we need rather than unrelated messages, and thus proceed in-depth analysis for disaster management is possible.
- Line 19: What is “school9 building”? Or is it just a typo?
Ans.
Yes, it is a typo. It should be a school building. We fixed it in the revised version.
- Line 210: What do you mean by: “The reference model adapted from the general Mandarin NER model developed by the open-source?].”?
Ans.
That is a missing reference. We fixed it in the revised version.
- There seems to be some missing references in text (see lines 140, 190, 211, 219, 221, 364, etc.)
Ans.
These are missing references. We fixed it in the revised version.
- Is it possible to generalize the algorithm for other languages, too? What would have to change?
Ans.
In the NLP area, there are some methods that try to solve this task, such as cross-lingual transfer [18], etc. However, it is not the main concern of this research because the application situation is mainly for Mandarin speakers. The generalization of other languages can be the future work of this research.

Reviewer 2 Report
As a paper on disaster management using data learning, it is considered to be a necessary study by analyzing the current situation and deriving problems in relation to the provision of disaster safety system support information. This research paper certainly has academic value. In this regard, The most important factor in this regard is the greatest justification for how to identify, collect, and provide disaster information.
I will list the review opinions as follows.
- The need for natural language processing (NLP) in disaster management needs to be addressed through various references. If complementary, the foundation of the study will be satisfied.
- Figure 5 must be modified. In the "start" and "end" process, it is necessary to explain more clearly why iteration is necessary through yes/no classification in the flow direction. If possible, it is a good idea to elaborate on the example flow.
Author Response
- The need for natural language processing (NLP) in disaster management needs to be addressed through various references. If complementary, the foundation of the study will be satisfied.
Ans.
We added two references in the introduction in the revised version. Safety Information Mining — What can NLP do in a disaster —; and Disaster response aided by tweet classification with a domain adaptation approach.
- Figure 5 must be modified. In the "start" and "end" process, it is necessary to explain more clearly why iteration is necessary through yes/no classification in the flow direction. If possible, it is a good idea to elaborate on the example flow.
Ans.
We added the example flow in the revised version.
